# Neither an Individualised Nor a Standardised Sodium Bicarbonate Strategy Improved Performance in High-Intensity Repeated Swimming, or a Subsequent 200 m Swimming Time Trial in Highly Trained Female Swimmers

**DOI:** 10.3390/nu16183123

**Published:** 2024-09-16

**Authors:** Josh W. Newbury, Matthew Cole, Adam L. Kelly, Lewis A. Gough

**Affiliations:** 1Research Centre for Life and Sport Science (CLaSS), School of Health Sciences, Birmingham City University, Birmingham B42 2LR, UK; adam.kelly@bcu.ac.uk (A.L.K.); lewis.gough@bcu.ac.uk (L.A.G.); 2Department of Sport, Hartpury University, Gloucestershire GL19 3BE, UK; matt.cole@hartpury.ac.uk

**Keywords:** sport nutrition, ergogenic aids, supplements, alkalosis, competition swimming

## Abstract

Inconsistent swimming performances are often observed following sodium bicarbonate (NaHCO_3_) ingestion, possibly because the time taken to reach peak blood buffering capacity is highly variable between individuals. Personalising NaHCO_3_ ingestion based on time-to-peak blood bicarbonate (HCO_3_^−^) could be a solution; however, this strategy is yet to be explored in swimming, or adequately compared to standardised NaHCO_3_ approaches. Therefore, six highly trained female swimmers ingested 0.3 g·kg BM^−1^ NaHCO_3_ in capsules to pre-determine their individual time-to-peak blood HCO_3_^−^. They then participated in three experimental trials, consisting of a 6 × 75 m repeated sprint swimming test, followed by a 200 m maximal time trial effort after 30 min active recovery. These experiments were conducted consuming a supplement at three different timings: individualised NaHCO_3_ (IND: 105–195 min pre-exercise); standardised NaHCO_3_ (STND: 150 min pre-exercise); and placebo (PLA: 90 min pre-exercise). Both NaHCO_3_ strategies produced similar increases in blood HCO_3_^−^ prior to exercise (IND: +6.8 vs. STND: +6.1 mmol·L^−1^, *p* < 0.05 vs. PLA) and fully recovered blood HCO_3_^−^ during active recovery (IND: +6.0 vs. STND: +6.3 mmol·L^−1^ vs. PLA, *p* < 0.05). However, there were no improvements in the mean 75 m swimming time (IND: 48.2 ± 4.8 vs. STND: 48.9 ± 5.8 vs. PLA: 49.1 ± 5.1 s, *p* = 0.302) nor 200 m maximal swimming (IND: 133.6 ± 5.0 vs. STND: 133.6 ± 4.7 vs. PLA: 133.3 ± 4.4 s, *p* = 0.746). Regardless of the ingestion strategy, NaHCO_3_ does not appear to improve exercise performance in highly trained female swimmers.

## 1. Introduction

Highly trained swimmers will typically complete 8–10 training sessions·week^−1^, many of which involve high-intensity exercise [1,2,3]. Such training is designed to stress the anaerobic energy systems, resulting in the intramuscular accumulation of hydrogen ions (H^+^) and inorganic phosphate (Pi) [4,5]. Indeed, during sustained high-intensity exercise, excessive H^+^ production overwhelms the natural blood bicarbonate (HCO_3_^−^) buffering mechanism, leading to the decline in blood and muscle pH [6]. Although it is contested that this acidic environment within the muscle causes exercise fatigue [7], it is under these conditions that other plausible fatiguing mechanisms occur, including a reduction in calcium ion (Ca^2+^) sensitivity and handling in the sarcoplasmic reticulum [8]; an impairment of key glycolytic enzymes [9]; and/or the depolarisation and inhibition of excitation–contraction coupling [10]. Thus, the ingestion of extracellular buffering agents, such as sodium bicarbonate (NaHCO_3_), could be an effective strategy to delay fatigue in key training sets, possibly enabling swimmers to make greater training adaptations over time.

Research has shown mixed performance outcomes when swimmers ingest NaHCO_3_ before repeated, high-intensity swimming exercise. For example, both Gao et al. [11] and Gough et al. [12] found NaHCO_3_ to improve swimming performance in the latter stages of 5 × 100 yard (0.7–0.9 s faster in final two bouts) and 8 × 50 m freestyle tests (0.5–1.3 s faster in final four bouts), respectively. However, these studies utilised regional/collegiate male swimmers, and therefore the results might not be applicable to swimmers of a higher training status. On the other hand, Zajac et al. [13] showed NaHCO_3_ to enhance the average swimming speed (+1.3%) of highly trained adolescent males across 4 × 50 m freestyle sprints. In contrast to the previous research, performance was only improved in the initial swimming bout (0.5 s faster) and not the later sprints when fatigue was expected to occur. Finally, Campos et al. [14] failed to find any NaHCO_3_ benefits towards 6 × 100 m freestyle time trials in highly trained male (*n* = 7) and female (*n* = 3) swimmers. However, this study appeared to administer NaHCO_3_ at a suboptimal time point given that a capsule ingestion method was used (60 min pre-exercise rather than 120–150 min pre-exercise) [15]. Indeed, a potential reason for the differing performance responses is that, following NaHCO_3_ ingestion, the time taken to reach peak buffering capacity is highly variable between individuals (i.e., 40–240 min post-ingestion) [16,17,18,19]. Together, this suggests that individualised NaHCO_3_ strategies might be required to achieve a consistent performance effect. Only Boegman et al. [16] have directly compared individualised and standardised NaHCO_3_ ingestion strategies on exercise performance to date, reporting a 2 s improvement in 2000 m rowing performance with the individualised approach. Although, as this research did not include a placebo/control condition, nor did it provide an indication of the possible effects on repeated high-intensity exercise, further research is still required to confirm the efficacy of individualised NaHCO_3_ strategies.

Sodium bicarbonate also carries the possibility of accelerating the recovery of acid-base balance after fatiguing exercise, achieving a full recovery within approximately 20–40 min [20]. This may have important practical implications since swimming competitions often demand consecutive races to be performed with only short rest periods (<40 min). Pierce et al. [21] and Pruscino et al. [22] have previously investigated the effects of NaHCO_3_ on repeated time trial swimming exercises (2 × 200 yards with 20 min recovery and 2 × 200 m with 30 min recovery, respectively) in elite male swimmers; however, only the latter study found a performance improvement (1.6 s faster than placebo). Similarly, both studies also employed inconsistent standardised NaHCO_3_ strategies (0.2 g∙kg BM^−1^ in a solution, 60 min pre-exercise vs. 0.3 g∙kg BM^−1^ in capsules, delivered in seven doses across 90 min, respectively), which could have mitigated the ergogenic potential for some swimmers. Therefore, the aim of this study was two-fold: (a) to compare the effectiveness of an individualised NaHCO_3_ ingestion strategy versus a standardised approach and a placebo on repeated swimming exercise; and (b) to assess acid-base recovery following these NaHCO_3_ ingestion methods to establish whether they can support a subsequent 200 m swimming time trial performance.

## 2. Materials and Methods

### 2.1. Participants

This study took place at a high-performance swimming club following the COVID-19 return to sport, where twelve swimmers (six male, six female) met the inclusion criteria (aged ≥ 16 years, nationally competitive at 200 m distances). All 12 swimmers underwent pre-experimental testing to determine their individual time-to-peak blood HCO_3_^−^ concentration. However, attrition occurred within the male group of swimmers, resulting in five withdrawing from the study. This high dropout rate was thought to be coincidental following COVID-19 and not because of the study design. Due to the time and financial restraints of this study, these participants could not be replaced [23]. As the remaining male was considered an ‘elite’ athlete [24], his data were not included in analyses due to significantly faster performance times compared to the female cohort.

This study therefore focused on six female swimmers (age: 18 ± 1 years, height: 1.73 ± 0.10 m, body mass: 67.4 ± 7.7 kg, World Aquatic points: 657 ± 56). Given this cohort’s consistency in performance (see Section 2.5), an a priori power calculation suggested that six participants were sufficient to identify moderate effect sizes (≥0.50) in repeated sprint swimming, whereas nine swimmers were needed to identify small effects (≥0.20) (input parameters: within–between interactions repeated measures analysis of variance, three groups, six measures, α = 0.05, β = 0.80, correspondence = 0.9; G*Power, v.3.1.9.4, Universität Düsseldorf, Düsseldorf, Germany). At the time of the study, all swimmers were in preparation for the national championships and were completing a swimming volume of 52.0 ± 6.6 km·week^−1^, meeting the criteria for ‘highly trained’ athletes [24]. Institutional ethical approval was granted prior to the study and written informed consent was received from all swimmers (and parents/guardians where appropriate).

### 2.2. Pre-Experimental Procedures

This study consisted of five research trials: one pre-determination of time-to-peak blood HCO_3_^−^, one familiarisation trial, and three experimental trials conducted in a single-blind, randomised, and crossover design. All trials took place at the swimming club’s training facilities (25 m pool). Swimmers were asked to follow their habitual nutrition intakes and timings prior to exercise, albeit with the avoidance of acute ergogenic supplements during the experimental period. These instructions were given for two reasons: (a) to increase external validity [25]; and (b) to potentially reduce the severity of gastrointestinal side-effects [15]. Swimmers were requested to send a photograph of all the food, fluid, and supplement items consumed to the researcher 24 h prior to the familiarisation trial. This information was used to calculate energy, macronutrient, and fluid intakes by the lead researcher, with the original photographs then re-sent to swimmers to facilitate dietary replication for experimental trials. Some swimmers reported the co-ingestion of creatine (*n* = 4) and/or beta-alanine (*n* = 2). However, as these had been consumed consistently for more than 24 weeks, these swimmers were permitted to participate in the study. This was to account for these supplements being commonplace in the diets of highly trained swimmers [26,27], because they would not influence acute blood buffering capacity [19] and would not have been expected to produce large performance enhancements after 24 weeks [28,29]. Therefore, any observed performance enhancement was likely to be the result of acute NaHCO_3_ ingestion.

### 2.3. Supplement Timings

Swimmers were told they would be ingesting a supplement at three different pre-exercise timings in an attempt to blind the experimental conditions. Individualised (IND) NaHCO_3_ was determined in accordance with previously described methods [19]. Briefly, swimmers attended the trial having nutritionally prepared for training. After engaging in five min of seated rest, swimmers gave a 70 μL capillary blood sample to be immediately analysed by a portable blood gas analyser (ABL9, Radiometer Medical, Copenhagen, Denmark). This determined the circulating values of HCO_3_^−^, pH, Ca^2+^, potassium (K^+^), sodium (Na^+^), and chloride (Cl^−^). A further 5 μL was drawn for the analysis of blood lactate (La^−^) (Lactate Pro 2, Arkray, Kyoto, Japan), which subsequently enabled the calculation of the apparent SID as per Lloyd [30]: K^+^ + Na^+^ + Ca^2+^ − Cl^−^ − La^−^. Following baseline measures, swimmers ingested 0.3 g∙kg BM^−1^ NaHCO_3_ in capsules and remained seated for a minimum of 180 min. Repeated blood samples were collected at 15 min intervals to identify a peak blood HCO_3_^−^ value. This testing determined the IND NaHCO_3_ strategies to be timed between 105 and 195 min before the initial exercise bout. As the lead researcher conducted time-to-peak testing and administered NaHCO_3_, they were not blinded from the results.

The standardised (STND) NaHCO_3_ ingestion strategy was based on the group mean time-to-peak blood HCO_3_^−^ in this cohort, which was determined to be 150 min pre-exercise. This timing did not coincide with any of the swimmers’ IND NaHCO_3_ timings and aligned with previous proposals for standardised ingestion timings with NaHCO_3_ capsules [15,18,19]. Both NaHCO_3_ strategies consisted of 0.3 g∙kg BM^−1^ NaHCO_3_ administered in hydroxypropyl methylcellulose capsules (~772 ± 30 mg∙capsule^−1^, size 00, Bulk, Colchester, UK).

Finally, a placebo (PLA) was to be ingested 90 min pre-exercise, as this timing also did not coincide with any IND timing and is within the recommended ingestion window for NaHCO_3_ ingestion [31]. The PLA consisted of an equimolar Na^+^ dose (sodium chloride: 0.21 g∙kg BM^−1^, ASDA, Leeds, UK) to offset any possible ergogenic effects of Na^+^ ingestion [32]. Additional cornflour (ASDA, Leeds, UK) was added to PLA capsules to replicate appearance and fullness.

### 2.4. Experimental Procedures

Swimmers were requested to arrive at the training facility 10–15 min before their pre-exercise NaHCO_3_ timing. Upon arrival, swimmers engaged in 5 min of seated rest before giving a capillary blood sample to determine baseline (BASE) blood acid-base variables, in accordance with the previously described methods. Gastrointestinal side-effects (nausea, flatulence, stomach cramp, belching, stomach ache, bowel urgency, diarrhoea, vomiting, stomach bloating) and perceived readiness to exercise (PRE) were also monitored at BASE, using 10 × 200 mm visual analogue scales (VAS). These scales were labelled to describe ‘no symptom’ on the left side and ‘extreme symptom’ on the right side, with the swimmer making a mark on the line to quantify their discomfort/motivation.

Repeat blood, gastrointestinal, and PRE measures were taken on five further occasions: before warming up (45 min pre-exercise: 45-PRE); after warming up (immediately pre-exercise: 0-PRE); after 6 × 75 m maximal swimming sprints (immediately post-exercise: 0-POST); after 30 min active recovery (30-POST); and after a follow-up 200 m swimming time trial (immediately post-exercise: POST-TT). At the end of each trial, the greatest score for each of the nine gastrointestinal side-effects was combined to give an aggregated gastrointestinal disturbance score for each condition.

### 2.5. Swimming Exercise Tests

Swimmers began a self-selected 40 min warm-up prior to the first swimming test. Although individual routines varied, swimmers typically completed 10 min of land-based activity (~3 min skipping, ~3–5 min mobility, ~3–5 min strength exercises) followed by a progressive intensity 30 min pool warm-up (~1000 m). Following the warm-up, swimmers were organised into swimming lanes ready to complete 6 × 75 m maximal effort sprints in their specialist swimming stroke (five freestyle, one butterfly, one breaststroke). Each swimming bout was competed at 150 s intervals, which typically resulted in 40–60 s exercise and 90–110 s passive rest. This was a commonly used swimming test within this cohort, who demonstrated high test–retest reliability for an average 75 m swimming time (coefficient of variation [CV]: 0.2–2.3%) and ‘excellent’ reproducibility over four attempts (intraclass correlation coefficient [ICC]: *r* = 0.997, *p* < 0.001) [33,34]. Two experienced swimming coaches manually timed each swimming bout, with the mean time used as the performance measure. The mean 75 m swimming time and individual 75 m bouts were all analysed for performance effects.

A 30 min active recovery period then ensued where swimmers completed 600–1000 m of low intensity swimming (~20 min), before engaging in foam rolling (~0–5 min) and/or passive rest (~5–10 min). Swimmers then completed a maximal 200 m swimming time trial from a dive start, using their specialist strokes. This distance was selected based on previous research by Pierce et al. [21] and Pruscino et al. [22]. Swimmers completed both exercise tests in their own swimming lanes, with a maximum of two swimmers completing the protocol at any one time. All warm-ups, swimming lanes, and recovery strategies were recorded and kept consistent for each trial. Swimmers were asked for their ratings of perceived exertion (RPE) after each swimming test, which were collected using a CR10 Borg scale [35].

### 2.6. Statistical Analysis

All statistical tests were carried out using SPSS (v.25, IBM, Chicago, IL, USA). Prior to analyses, all data were screened for normality and homogeneity of variance/sphericity using the Shapiro–Wilk and Mauchly tests, respectively. A one-way repeated measures analysis of variance (ANOVA) test was used to compare the swimming performances (mean 75 m time, 200 m time trial), RPE, and aggregated gastrointestinal side-effects between the three experimental conditions (IND vs. STND vs. PLA). Six (BASE, 45-PRE, 0-PRE, 0-POST, 30-POST, POST-TT) x three (IND, STND, PLA) repeated measures ANOVA tests were conducted to compare differences in HCO_3_^−^, SID, Na^+^, K^+^, Ca^2+^, Cl^−^, La^−^, and PRE across the study timeframe, as well to compare performance in the individual bouts of the 6 × 75 m swimming test. If sphericity was violated in the ANOVA tests, then the degrees of freedom and *p* values were adjusted using the appropriate Huyn–Feldt (epsilon value > 0.75) or Greenhouse–Geiser (epsilon value < 0.75) corrections. Where main effects or interactions were observed, partial eta squared (*Pη*^2^) effect sizes were reported and post hoc pairwise comparisons were determined via the Bonferroni correction. The effect sizes for *Pη*^2^ were interpreted as ‘small’ (0.01–0.05), ‘moderate’ (0.06–0.13), and ‘large’ (≥0.14) [36]. The effect sizes for pairwise comparisons were calculated using the Hedge’s *g* bias correction, which accounted for the bias in Cohen’s *d* with small sample sizes (*n* < 20) [37]. These effect sizes (*g*) were interpreted as ‘trivial’ (≤0.19), ‘small’ (0.20–0.49), ‘moderate’ (0.50–0.79), and ‘large’ (≥0.80) [36]. The smallest worthwhile changes (SWC) of 1.1 s (6 × 75 m) and 1.6 s (200 m time trial) were calculated by multiplying the standard deviation of this cohort’s previous performance data by 0.2, in accordance with Bernards et al. [38]. The CV was calculated by dividing the standard deviation of the data by the mean and multiplying by 100 [33]. All data are presented as the mean ± standard deviation.

## 3. Results

### 3.1. Blood Metabolites

Initial time-to-peak testing found that all six swimmers achieved an absolute increase in blood HCO_3_^−^ of more than +5 mmol·L^−1^ (mean: +7.7 ± 1.1 mmol·L^−1^, CV = 14%), albeit at variable post-ingestion time points (137 ± 40 min, CV = 29%). During the experimental trials, elevated blood HCO_3_^−^ concentrations were identified following IND and STND NaHCO_3_ ingestion (*p* = 0.024, *Pŋ*^2^ = 0.49, Figure 1). Specifically, blood HCO_3_^−^ was elevated at the pre-exercise time points of 0-PRE (IND vs. PLA: +6.8 mmol·L^−1^, *p* = 0.004, *g* = 3.38; STND vs. PLA: +6.1 mmol·L^−1^, *p* = 0.004, *g* = 3.15) and 30-POST (IND vs. PLA: +6.0 mmol·L^−1^, *p* = 0.013, *g* = 3.38; STND vs. PLA: +6.3 mmol·L^−1^, *p* = 0.002, *g* = 3.15). No differences were found between the two NaHCO_3_ conditions at any time point (all *p* > 0.05).

While blood HCO_3_^−^ was found to be increased with NaHCO_3_ at the group mean level, not all swimmers achieved a +5 mmol·L^−1^ increase prior to exercise. This included variable blood responses to supplementation at 0-PRE compared to PLA (IND: mean: +5.2 ± 2.7 mmol·L^−1^, range: +0.1 to 7.2 mmol·L^−1^; STND: mean: +4.5 ± 2.0 mmol·L^−1^, range: +1.8 to 7.4 mmol·L^−1^), as well as at 30-POST (IND: mean: +3.4 ± 2.8 mmol·L^−1^, range: –0.8 to +6.9 mmol·L^−1^; STND: mean: +3.8 ± 2.2 mmol·L^−1^, range: +1.1 to 6.8 mmol·L^−1^).

No differences in the apparent SID occurred between the two NaHCO_3_ strategies nor the PLA treatment across the study timeframe (*p* = 0.761, *Pŋ*^2^ = 0.11; Figure 2). Despite no statistical significance, a comparison of STND to PLA produced large and moderate effect sizes at 0-PRE (+4.0 mEq·L^−1^, *g* = 0.86) and 30-POST (+3.2 mEq·L^−1^, *g* = 0.70), respectively. Yet, this same effect was not present with the IND approach (0-PRE vs. PLA: +0.5 mEq·L^−1^, *g* = 0.08; 30-POST vs. PLA: +1.0 mEq·L^−1^, *g* = 0.22).

Statistically significant differences were not identified between conditions for Na^+^ (*p* = 0.358, *Pŋ*^2^ = 0.19), K^+^ (*p* = 0.280, *Pŋ*^2^ = 0.22), Cl^−^ (*p* = 0.089, *Pŋ*^2^ = 0.37), or Ca^2+^ (*p* = 0.157, *Pŋ*^2^ = 0.29). However, large effect sizes were observed within each of the measured variables (Figure 3). When comparing both IND and STND NaHCO_3_ strategies versus PLA, large effect sizes occurred at every post-ingestion time point for K^+^, Cl^−^, and Ca^2+^ (all *g* > 0.80). For Na^+^, large effect sizes occurred post-exercise only with STND NaHCO_3_ ingestion (30-POST vs. PLA: *g* = 0.88; POST-TT vs. PLA: *g* = 0.82), with moderate effect sizes observed for the IND approach (0-POST vs. PLA: *g* = 0.72; 30-POST vs. PLA: *g* = 0.72).

No differences in blood La^−^ concentrations were observed between all three groups throughout the investigation (*p* = 0.223, *Pŋ*^2^ = 0.26, Figure 4). Moderate effect sizes were identified when comparing both post-exercise values between the IND and PLA conditions (0-POST: +3.0 mmol·L^−1^, *g* = 0.68; POST-TT: +3.7 mmol·L^−1^, *g* = 0.77), but only following POST-TT when comparing the STND versus PLA conditions (0-POST: +3.0 mmol·L^−1^, *g* = 0.73).

### 3.2. Swimming Performance

No differences were observed in the mean 75 m swimming time (*p* = 0.302, *Pŋ*^2^ = 0.21) or any individual swimming bout (*p* = 0.529, *Pŋ*^2^ = 0.14) during the 6 × 75 m test (Table 1). This included four of six swimmers producing highly repeatable swimming times across the experimental trials (mean 75 m time: ±0.1–0.7 s, CV = 0.1–0.9%). The other two swimmers both recorded mean swimming times above the SWC (±1.1 s). There was an improvement in participant three when ingesting NaHCO_3_ (IND: 50.3 ± 0.6 s and STND: 51.3 ± 0.7 s vs. PLA: 53.9 s ± 0.4 s), and a decrement in participant four when ingesting STND NaHCO_3_ (IND: 57.1 ± 0.5 s and PLA: 57.2 ± 0.2 s vs. STND: 59.7 ± 0.3 s). The effect sizes for the mean 75 m swim time, each 75 m swimming bout, and aggregated time-to-complete the 6 × 75 m swimming test were all trivial (*g* < 0.20).

Neither group could also be differentiated in the 200 m swimming time trial performance (IND: 133.6 ± 5.0 s, STND: 133.6 ± 4.7 s, PLA: 133.3 ± 4.4 s; *p* = 0.746, *Pŋ*^2^ = 0.03; all *g* < 0.20). Similarly, repeatable performances were observed in five of six swimmers (±0.4–1.6 s, CV = 0.2–0.6%), with participant four exceeding the SWC (±1.6 s) with a slower swimming time following IND NaHCO_3_ ingestion (IND: 140.7 s vs. STND: 137.9 s and PLA: 136.9 s).

### 3.3. Perceptual Measures

There were no differences in the RPE reported between conditions following the 6 × 75 m swimming test (IND: 9.3 ± 0.6 vs. STND: 9.2 ± 0.8 vs. PLA: 8.5 ± 1.5 units; *p* = 0.277, *Pŋ*^2^ = 0.23) or the 200 m time trial performance (IND: 9.0 ± 1.2 vs. STND: 9.0 ± 0.9 vs. PLA: 9.0 ± 1.2 units; *p* = 0.751, *Pŋ*^2^ = 0.06). The perceived readiness to exercise was also no different between all three conditions across the study timeframe (*p* = 0.643, *Pŋ*^2^ = 0.10), which peaked immediately before the 6 × 75 m swimming test (0-PRE scores, IND: 7.0 ± 1.9 units, STND: 7.3 ± 1.4 units, PLA: 6.7 ± 1.7 units, all *g* < 0.50). The perceived readiness to exercise scores before the 200 m freestyle time trial were also trivial between conditions (IND: 6.0 ± 2.3 vs. STND: 5.8 ± 3.4 vs. PLA: 5.6 ± 2.5 units; all *g* < 0.20).

### 3.4. Gastrointestinal Side-Effects

The aggregated scores for gastrointestinal side-effects did not differ between supplemental conditions (*p* = 0.338, *Pŋ*^2^ = 0.20), with the mean scores being 23.5 ± 16.1 units (IND), 15.7 ± 10.5 units (STND), and 21.8 ± 18.2 units (PLA). These scores were highly variable, with large ranges in aggregated scores reported for IND (3–51 units, CV = 69%), STND (4–34 units, CV = 67%), and PLA (6–47 units, CV = 84%). The most severe gastrointestinal side-effects reported by individuals in each condition are presented in Table 2.

### 3.5. Order Effects and Supplement Predictions

No order effects were identified between trials for the mean 75 m swimming time (*p* = 0.767, *Pŋ*^2^ = 0.04) or 200 m freestyle time trial performances (*p* = 0.265, *Pŋ*^2^ = 0.20). Moreover, swimmers were successfully blinded in this study, only correctly predicting whether they consumed either NaHCO_3_ (IND or STND) or PLA on 33% of occasions.

## 4. Discussion

The primary purpose of this investigation was to assess the effect of an individualised versus a standardised NaHCO_3_ ingestion strategy on repeated, high-intensity swimming performance. Despite both NaHCO_3_ strategies enhancing blood HCO_3_^−^ concentrations prior to exercise, neither provided an ergogenic benefit compared to a Na^+^-matched placebo. The secondary purpose was to observe acid-base recovery and whether this could improve performance in a follow-up 200 m swimming time trial. After 30 min active recovery, both NaHCO_3_ strategies recovered blood HCO_3_^−^ concentration to the elevated levels observed prior to exercise, although this again failed to produce any ergogenic benefits. These results infer that NaHCO_3_, regardless of dosing strategy, may not be an effective strategy for highly trained female swimmers to improve sprint swimming performances in training, or enhance recovery for subsequent swimming time trial bouts in competition.

Individualising NaHCO_3_ ingestion did not produce greater pre-exercise blood HCO_3_^−^ concentrations compared a standardised approach (+0.7 mmol·L^−1^, *g* = 0.26), supporting previous research in world-class rowers (+0.5 mmol·L^−1^, *g* = 0.29) [16]. However, the previous study did observe pre-exercise blood HCO_3_^−^ increases of +6 mmol·L^−1^ (individualised) and +5.5 mmol·L^−1^ (standardised) with both NaHCO_3_ strategies [16], which have been associated with ‘almost certain’ and ‘possible’ ergogenic benefits, respectively [39]. In the present study, blood HCO_3_^−^ increases did not exceed the proposed ergogenic threshold of +5 mmol·L^−1^ [39], despite all swimmers reaching this threshold in initial time-to-peak testing. This was likely due to the differing dietary controls between studies, with the athletes in Boegman et al. [16] consuming a standardised snack three hours before NaHCO_3_ ingestion and staying fasted for ~4–6 h before exercise. Although, this meal pattern is unlikely to be followed in applied practice. In contrast, this study encouraged swimmers to follow their normal dietary practices with NaHCO_3_ being supplemental to their preparation. This resulted in swimmers consuming various meal patterns (e.g., a large meal, two small meals, snacking) and macronutrient amounts (e.g., 35–180 g carbohydrate, 21–68 g protein, 7–59 g fat), either alongside or following NaHCO_3_ ingestion. These dietary differences could have therefore slowed NaHCO_3_ absorption characteristics [19,40,41], and highlight a key flaw when attempting to individualise NaHCO_3_ in practice. As such, in order to truly individualise NaHCO_3_ supplementation outside of laboratory settings, athletes may have to undergo time-to-peak testing multiple times, firstly to identify the fasted blood HCO_3_^−^ time course, and then to observe how different meals, snacks, and drinks impact their pharmacokinetics. Although, this presents a considerable time and financial burden for athletes and their sports teams.

Alternatively, the importance of achieving a +5 mmol·L^−1^ increase in blood HCO_3_^−^ is questionable. Previous studies in swimmers have demonstrated ergogenic benefits of NaHCO_3_ when pre-exercise increases in blood HCO_3_^−^ were only +3.5–4.4 mmol·L^−1^ above baseline values [11,13,42], which were in accordance with the observations in this study (IND: +4.9 mmol·L^−1^, STND: +4.1 mmol·L^−1^). This adds to the premise that an increased blood buffering capacity might not be the primary ergogenic mechanism following NaHCO_3_ ingestion [7], and instead the altered SID could delay muscle depolarisation and maintain excitation–contraction coupling during high-intensity exercise [43]. In this study, large effect sizes might have indicated that circulating levels of K^+^, Cl^−^, and Ca^2+^ had decreased following NaHCO_3_ ingestion, potentially signaling their intramuscular uptake [10,44]. However, the shifts in strong ions, as well as the collective SID, did not reach statistical significance versus the placebo treatment, which might explain the lack of ergogenic effects in this study compared to others [20,45,46]. On the other hand, the Na^+^-matched placebo contributed towards similar gastrointestinal discomfort to NaHCO_3_ ingestion, perhaps due to acute mucosal irritation and fluid shifts in the stomach [47]. This could also suggest that the side-effects from NaHCO_3_ and the placebo were equally ergogenic or ergolytic for performance, warranting the use of non-supplemental control conditions in the future to elucidate any possible effects of NaHCO_3_ on swimming performance.

Another reason why NaHCO_3_ may have been ineffective was due to the exercise protocols. The first swimming exercise involved 6 × 75 m maximal effort bouts with short rest periods, which was selected based on its familiarity and repeatability in this cohort. This test induced a large acid-base perturbation, which was evidenced through declines in blood HCO_3_^−^ (−9–15 mmol·L^−1^), apparent SID (−11–15 mEq·L^−1^), and pH (−0.19–0.21 units; data not presented) immediately after exercise. However, because such large perturbations were produced over a short timeframe (<60 s), the rapid rates of change in pH (and thus intramuscular H^+^ accumulation) were likely to outweigh the possible ergogenic mechanisms expected from NaHCO_3_ supplementation [48,49,50]. The second exercise protocol was a 200 m swimming time trial distance that was expected to be enhanced with NaHCO_3_ ingestion [51]. Similarly, no performance benefits were observed following either NaHCO_3_ ingestion strategy, despite both conditions recovering and increasing blood HCO_3_^−^ concentrations in the 30 min recovery window (both +3.5 mmol·L^−1^ vs. baseline; +6 mmol·L^−1^ vs. PLA). Although, while these blood values represented an elevated blood buffering capacity prior to exercise, this may not have contributed to an ergogenic effect because of the following: (a) there were little differences in the SID between the NaHCO_3_ and PLA conditions; and/or (b) the initial repeated sprint exercise impeded exercise by producing muscle damage and fatigue at the neuromuscular level [52,53]. Thus, as swimming competitions would not feature exhausting repeated sprints before a race, further research is needed to investigate the effects of individualised NaHCO_3_ prior to two time trial efforts. Moreover, it is also suspected that highly trained swimmers might also complete 200 m distances at intensities that also produce a rapid rate of pH change. As such, further swimming distances that require a sustained effort should also be investigated, particularly 400–800 m distances as these are of similar duration to 4 km time trial cycling (~4–9 min), which appears to receive a consistent NaHCO_3_ benefit [20,45,54].

Due to the financial and time burdens of this study, it was not possible to replace the male participants who withdrew during data collection. This left the study with a sample size of just six female participants. While the consistency of this cohort meant that this was sufficient to identify medium effect sizes, it is a concern that type II statistical errors were possible, and marginal improvements associated with ergogenic aids were overlooked [55]. Nonetheless, this sample size was in line with previous NaHCO_3_ research (*n* = 6–10) [11,13,14,21,22,56,57,58], and data were presented at the individual level using a SWC, in which no trends for a NaHCO_3_ benefit could be established. Another limitation was that menstrual cycle stage was not considered within this female cohort. Despite there being little information to suggest that menstrual cycle stage affects physical performance or physiological responses to NaHCO_3_ supplementation in highly trained female athletes [59,60], it is recognised that the current assumption was based on limited research [61], and was not specific to swimmers. Indeed, it is possible that hydration (which can be affected by Na^+^ ingestion) and gastrointestinal symptoms can be influenced by the menstrual cycle [62,63], potentially impairing the ergogenic response to NaHCO_3_^−^ ingestion. Further research involving female athletes is therefore a necessity to better understand how nutrition and physiology are affected by nutritional ergogenic aids.

## 5. Conclusions

Highly trained female swimmers ingesting NaHCO_3_ capsules at either an individualised (105–195 min) or standardised (150 min) pre-exercise time point received no ergogenic benefit for repeated swimming sprints, nor a follow-up 200 m time trial after 30 min active recovery. Furthermore, the time consuming and expensive approach of individualising NaHCO_3_ timings did not provide greater pre-exercise blood HCO_3_^−^ responses compared to a standardised approach at 150 min before exercise, questioning this supplement strategy in practice. One reason might be because the current method of identifying time-to-peak blood HCO_3_^−^ takes place in a postprandial state, which differs from how swimmers ingest NaHCO_3_ for training and competitions (i.e., not accounting for pre-exercise snacks and drinks). This is an important logistical consideration that requires further research to optimise NaHCO_3_ supplementation. It is also speculated that the exercise tests used in this study were too short and intense to benefit from an increased rate of H^+^ removal; thus, it is plausible that NaHCO_3_ could be more effective in swimming exercise lasting between 4 and 9 min (400–800 m time trials). However, as there is a dearth of evidence involving middle-to-long distance swimming, this cannot be confirmed at present.

## Figures and Tables

**Figure 1 nutrients-16-03123-f001:**
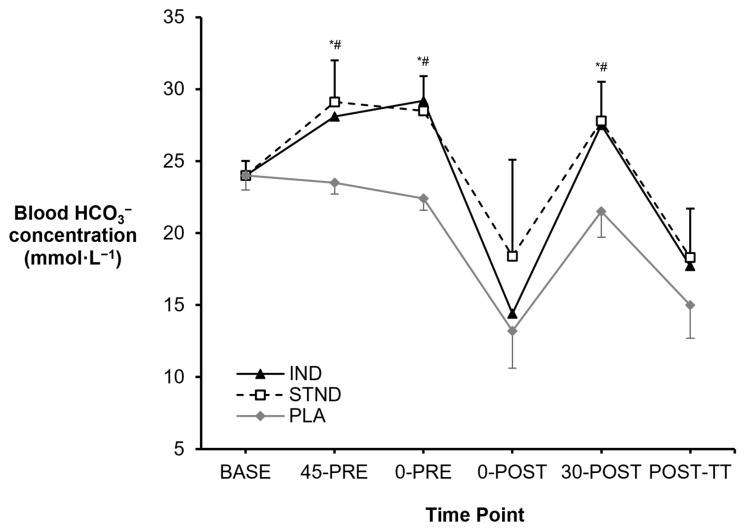
Changes in blood HCO_3_^−^ concentration observed across the study timeframe. * = IND different to PLA (*p* < 0.05). # = STND different to PLA (*p* < 0.05).

**Figure 2 nutrients-16-03123-f002:**
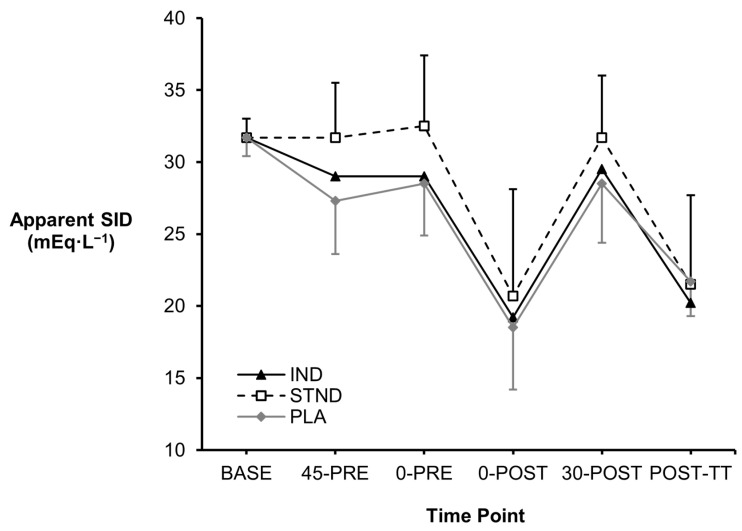
Changes in the apparent SID observed across the study timeframe.

**Figure 3 nutrients-16-03123-f003:**
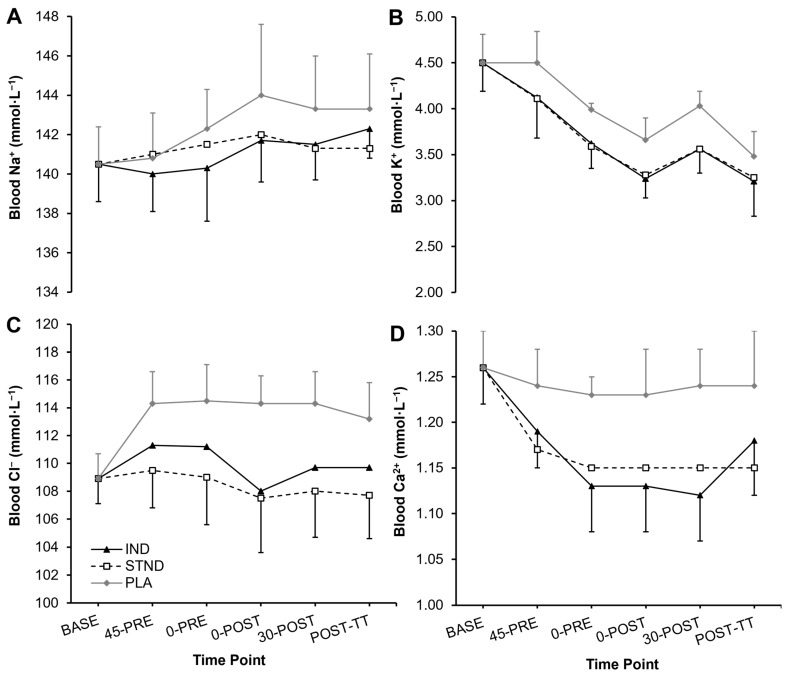
Changes in the blood values of (**A**) sodium (Na^+^), (**B**) potassium (K^+^), (**C**) chloride (Cl^−^), and (**D**) calcium (Ca^2+^) across the study timeframe.

**Figure 4 nutrients-16-03123-f004:**
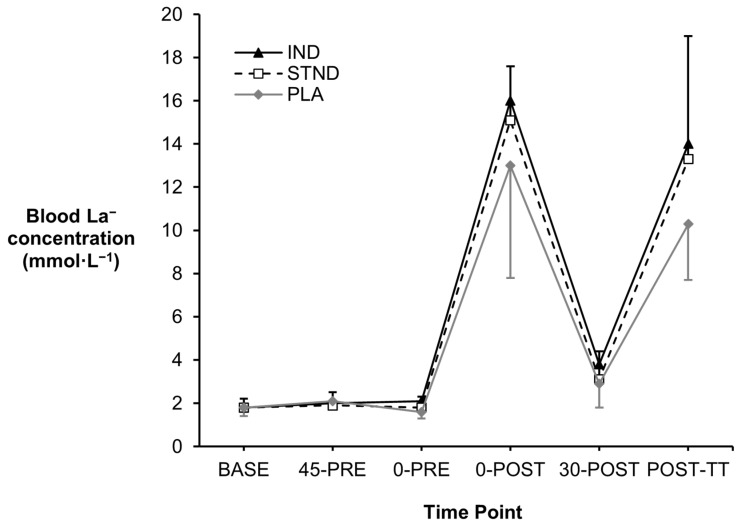
Changes in blood La^−^ concentration observed across the study timeframe.

**Table 1 nutrients-16-03123-t001:** Mean and aggregated performance times in the 6 × 75 m swimming test.

Performance Variable	Ingestion Strategy		
	IND	STND	PLA
Bout 1 (s)	48.5 ± 4.7	48.9 ± 5.4	49.1 ± 4.9
Bout 2 (s)	48.0 ± 4.6	48.7 ± 5.7	49.0 ± 5.1
Bout 3 (s)	48.2 ± 4.9	48.8 ± 5.8	49.1 ± 5.3
Bout 4 (s)	47.9 ± 4.8	48.6 ± 5.9	49.0 ± 5.2
Bout 5 (s)	48.4 ± 4.6	49.0 ± 6.1	49.3 ± 5.0
Bout 6 (s)	48.2 ± 5.1	49.2 ± 6.0	49.1 ± 5.4
Mean 75 m (s)	48.2 ± 4.8	48.9 ± 5.8	49.1 ± 5.1
Mean aggregated (s)	289.2 ± 28.6	293.3 ± 34.8	294.5 ± 30.9

Mean ± standard deviation.

**Table 2 nutrients-16-03123-t002:** Most severe gastrointestinal side-effects reported by swimmers in each trial.

Swimmer	Ingestion Strategy		
	IND	STND	PLA
**1**	Stomach Ache 4.9/10 (0-POST)	Stomach Bloating 5/10 (30-POST)	Nausea 3.5/10 (0-PRE)
**2**	Stomach Ache 4.1/10 (0-PRE)	Nausea 3.2/10 (45-PRE)	Vomiting 10/10 (45-PRE)
**3**	Nausea 6.2/10 (0-POST)	Bowel Urgency 1.7/10 (0-POST)	Bowel Urgency 1.5/10 (POST-TT)
**4**	Stomach Ache 0.6/10 (45-PRE)	Stomach Ache 2.4/10 (BASE)	Nausea 8.2/10 (45-PRE)
**5**	Nausea 8.9/10 (0-PRE)	Nausea 7/10 (45-PRE)	Nausea 8.7/10 (45-PRE)
**6**	Nausea 10/10 (POST-TT)	Nausea 6.6/10 (30-POST)	Nausea 6/10 (0-PRE)

## Data Availability

The original data contributions in the study are included in the article, further inquiries can be directed to the corresponding author.

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
