# Peer review of "Neither an Individualised Nor a Standardised Sodium Bicarbonate Strategy Improved Performance in High-Intensity Repeated Swimming, or a Subsequent 200 m Swimming Time Trial in Highly Trained Female Swimmers"

_nutrients, 2024, doi:10.3390/nu16183123_

Round 1

Reviewer 1 Report

Comments and Suggestions for Authors

Dear Authors,

I would like to commend you on the thoroughness and depth of your research. The investigation into the effects of individualized versus standardized sodium bicarbonate ingestion strategies on swimming performance is both timely and relevant, particularly given the growing interest in personalized nutrition strategies in sports. The study design, particularly the crossover format, and the detailed analysis of blood biomarkers provide a robust foundation for the findings presented. Nevertheless, upon thorough examination, it is evident that the manuscript would greatly benefit from substantial revisions to address several methodological and analytical concerns. The following section provides detailed feedback to guide the necessary revisions.

I. STATISTICAL ANALYSIS
- General Linear Models (GLM) vs. Analysis of Variance (ANOVA):
The manuscript employs two types of ANOVA to analyze the data from the repeated measures design. Although ANOVA is a widely accepted method for comparing means across conditions, it may not fully account for the within-subject variability inherent in the study design, in which the same athletes were exposed to multiple conditions. A General Linear Model (GLM) may offer greater flexibility, allowing for the inclusion of both fixed effects (e.g., treatment conditions) and random effects (e.g., individual differences among athletes). Furthermore, GLM is better equipped to handle the repeated measures aspect of the data, potentially providing a more accurate reflection of the treatment effects.
It is recommended that you re-examine the use of GLM for your analysis or provide a comprehensive justification for the selection of ANOVA over GLM. This justification should encompass an examination of how your chosen method effectively addresses within-subject variability and how this decision affects the interpretation of your results and the estimation of effect sizes.

II. SAMPLE SIZE AND STATISTICAL POWER
- Sample Size Considerations:
The relatively modest sample size of six participants, while comparable to that of some previous studies, constrains the statistical power of the findings. This is a matter of particular concern, given the considerable variability in individual responses to sodium bicarbonate ingestion that your study has identified. While the use of effect sizes is praiseworthy and helps contextualize the findings, the low sample size necessitates a cautious interpretation of the results.
If feasible, it would be beneficial to discuss the implications of this small sample size in greater detail, including any potential biases or limitations it introduces. It may also be beneficial to conduct a power analysis to ascertain the adequacy of the sample size for detecting meaningful effects.

III. METHODOLOGICAL TRANSPARENCY
- Dietary Control and Timing:
Your discussion highlights the potential challenges of controlling dietary intake and the timing of sodium bicarbonate ingestion, which could have influenced the outcomes. However, it would be beneficial to gain a deeper understanding of the impact these factors have had on the findings. Given that individualized supplementation strategies are a key focus of your study, it would be helpful to clarify how these logistical challenges were addressed and how they might have affected the results.
It would be valuable to provide more detail on how dietary intake and timing were controlled or accounted for in your analysis. If any variations could not be controlled, discussing their potential impact on the results could enhance the transparency and credibility of your study.

IV. BROADER IMPLICATIONS AND FUTURE DIRECTIONS
- Interpretation of Findings:
Your findings suggest that, regardless of the ingestion strategy, sodium bicarbonate may not provide an ergogenic benefit in the context you studied. This is a valuable contribution to the field, but it might be beneficial to discuss the broader implications of these results in more depth. For instance, might these findings potentially influence future research on ergogenic aids in sports, or could any recommendations be made for practitioners based on your study?
I encourage you to expand on the discussion of the implications of your findings, particularly how they might inform future research or practical applications in sports nutrition. In addition, it would be valuable to acknowledge the limitations of the study and suggest potential avenues for further investigation.

In summary, your manuscript provides valuable insight into the effects of sodium bicarbonate on swimming performance. However, significant revisions are needed to address the methodological and analytical concerns outlined above. By providing a more detailed rationale for your statistical choices, addressing sample size limitations, and improving the clarity of your reporting, your manuscript will be strengthened and better positioned to contribute to the ongoing discourse in sports nutrition research.

Thank you for your contributions to this important area of research. I look forward to receiving the revised version of your manuscript.

Comments on the Quality of English Language

Overall, the manuscript is well-written with clear and concise language. The authors have demonstrated a commendable command of the English language, with only a few minor errors in grammar, punctuation, or syntax. On the other hand, there are a few areas that could benefit from some improvement.
1. It would be beneficial to ensure that the terminology is consistent throughout the manuscript. As a suggestion, it might be helpful to use terms like "pre-exercise" and "pre-experimental" consistently to avoid any confusion.
2. It might be helpful to consider shortening some sentences, as they are quite lengthy and could be difficult to follow. It might be helpful to break these into shorter sentences to enhance clarity. For instance, the sentence in the introduction that discusses the training regimen of swimmers could be rephrased to make it easier to read.
3. It might be beneficial to consider balancing the use of the passive voice with the active voice, as it is appropriate in scientific writing, to maintain reader engagement. It might be helpful to consider revising some passive constructions to active voice where appropriate.
4. There are a few minor grammatical errors, such as missing commas or misplaced modifiers, which could be improved. It might be helpful to have someone proofread the text to catch these issues. For instance, in some instances, it might be helpful to include a comma after introductory phrases for better readability (e.g., "In summary, ingesting NaHCO3 capsules...").
5. It would be beneficial to maintain consistency in verb tenses, especially when describing the study's methodology and findings. There are a few instances where the tense shifts unexpectedly, which could potentially disrupt the flow.

While the quality of English in the manuscript is exemplary, there may be a few minor areas for enhancement to further elevate the clarity and readability. With these adjustments suggested, the manuscript will be well-suited for publication.

Author Response

  1. STATISTICAL ANALYSIS
    - General Linear Models (GLM) vs. Analysis of Variance (ANOVA):
    The manuscript employs two types of ANOVA to analyze the data from the repeated measures design. Although ANOVA is a widely accepted method for comparing means across conditions, it may not fully account for the within-subject variability inherent in the study design, in which the same athletes were exposed to multiple conditions. A General Linear Model (GLM) may offer greater flexibility, allowing for the inclusion of both fixed effects (e.g., treatment conditions) and random effects (e.g., individual differences among athletes). Furthermore, GLM is better equipped to handle the repeated measures aspect of the data, potentially providing a more accurate reflection of the treatment effects.
    It is recommended that you re-examine the use of GLM for your analysis or provide a comprehensive justification for the selection of ANOVA over GLM. This justification should encompass an examination of how your chosen method effectively addresses within-subject variability and how this decision affects the interpretation of your results and the estimation of effect sizes.

Response: To our knowledge, the procedure for calculating the difference between three means (i.e., our IND vs. STND vs. PLA data) is to perform a repeated measures ANOVA. Upon further exploration of GLMs, I was directed to perform the following procedure in SPSS when considering our data: ‘General Liner Models’ > ‘Repeated Measures’, which was the statistical approach that was used in this manuscript and other similar studies (de Salles Painelli et al., 2013; Esen et al., 2022; Pruscino et al., 2008). Other GLM approaches tended to require other co-variates, such as qualitative groupings.

We were also aware that high intra-individual variability would be present in our results and attempted to counter this by presenting the mean and standard deviation, as well as the coefficient of variation or data range. We also compared individual performance data in relation to a smallest worthwhile change rather than focussing on group mean values. However, we would be willing to perform the recommended GLM test with further direction.

2. SAMPLE SIZE AND STATISTICAL POWER
- Sample Size Considerations:
The relatively modest sample size of six participants, while comparable to that of some previous studies, constrains the statistical power of the findings. This is a matter of particular concern, given the considerable variability in individual responses to sodium bicarbonate ingestion that your study has identified. While the use of effect sizes is praiseworthy and helps contextualize the findings, the low sample size necessitates a cautious interpretation of the results.
If feasible, it would be beneficial to discuss the implications of this small sample size in greater detail, including any potential biases or limitations it introduces. It may also be beneficial to conduct a power analysis to ascertain the adequacy of the sample size for detecting meaningful effects.

Response: We have now included a power calculation in lines 100 – 107 that explains that six participants would be sufficient to identify moderate effect sizes, yet nine were required to identify small effect sizes. We acknowledge that our small sample size could have contributed to type II errors and a possibility that  small effect sizes could have been overlooked, which is problematic considering that ergogenic aids are associated with marginal improvements in performance. Nonetheless, we thank the reviewer for appreciating our use of effect sizes and individual results, as our study can still provide a worthwhile contribution to sodium bicarbonate research.

III. METHODOLOGICAL TRANSPARENCY
- Dietary Control and Timing:
Your discussion highlights the potential challenges of controlling dietary intake and the timing of sodium bicarbonate ingestion, which could have influenced the outcomes. However, it would be beneficial to gain a deeper understanding of the impact these factors have had on the findings. Given that individualized supplementation strategies are a key focus of your study, it would be helpful to clarify how these logistical challenges were addressed and how they might have affected the results.
It would be valuable to provide more detail on how dietary intake and timing were controlled or accounted for in your analysis. If any variations could not be controlled, discussing their potential impact on the results could enhance the transparency and credibility of your study.

Response: We decided to allow swimmers to consume their normal nutritional intake to make the findings more applicable to athletes. For instance, most studies either standardise a meal or investigate sodium bicarbonate in a fasted state, whereas in reality, athletes would consume their normal nutrition and use sodium bicarbonate as a supplement to their preparation. We have therefore acknowledged that this causes a logistical issue when using the current methods to individualise sodium bicarbonate in practice, and as a result, the current methods for individualising supplementation are likely to be flawed (lines 340 – 355). Our results show that, to truly individualise sodium bicarbonate supplementation, athletes may have to undergo multiple time-to-peak tests to firstly understand their fasted HCO3 time course, before then re-assessing the time course under different dietary conditions as athletes would snack and drink in the 30 – 60 min before exercise (lines 430 – 433).

IV. BROADER IMPLICATIONS AND FUTURE DIRECTIONS
- Interpretation of Findings:
Your findings suggest that, regardless of the ingestion strategy, sodium bicarbonate may not provide an ergogenic benefit in the context you studied. This is a valuable contribution to the field, but it might be beneficial to discuss the broader implications of these results in more depth. For instance, might these findings potentially influence future research on ergogenic aids in sports, or could any recommendations be made for practitioners based on your study?
I encourage you to expand on the discussion of the implications of your findings, particularly how they might inform future research or practical applications in sports nutrition. In addition, it would be valuable to acknowledge the limitations of the study and suggest potential avenues for further investigation.

Response: Thank you for pointing this out, the authors agree that we have overlooked adding avenues for further investigation, and have included these at the end of each paragraph in the discussion section. This includes investigation of blood bicarbonate changes following the intake of different meals (lines 351– 355), investigating repeated time-trials as these better replicate a competition scenario (lines 394 – 397), exploring the effects on longer time trial distances that have a more sustained high intensity component (lines 398 – 402), and the need for more research involving female athletes (lines 420 – 422).

In summary, your manuscript provides valuable insight into the effects of sodium bicarbonate on swimming performance. However, significant revisions are needed to address the methodological and analytical concerns outlined above. By providing a more detailed rationale for your statistical choices, addressing sample size limitations, and improving the clarity of your reporting, your manuscript will be strengthened and better positioned to contribute to the ongoing discourse in sports nutrition research.

Response – Thank you for your comments, we believe the amendments we have now made as a result have significantly strengthening our manuscript.

Comments on the Quality of English Language

Overall, the manuscript is well-written with clear and concise language. The authors have demonstrated a commendable command of the English language, with only a few minor errors in grammar, punctuation, or syntax. On the other hand, there are a few areas that could benefit from some improvement.
1. It would be beneficial to ensure that the terminology is consistent throughout the manuscript. As a suggestion, it might be helpful to use terms like "pre-exercise" and "pre-experimental" consistently to avoid any confusion.
2. It might be helpful to consider shortening some sentences, as they are quite lengthy and could be difficult to follow. It might be helpful to break these into shorter sentences to enhance clarity. For instance, the sentence in the introduction that discusses the training regimen of swimmers could be rephrased to make it easier to read.
3. It might be beneficial to consider balancing the use of the passive voice with the active voice, as it is appropriate in scientific writing, to maintain reader engagement. It might be helpful to consider revising some passive constructions to active voice where appropriate.
4. There are a few minor grammatical errors, such as missing commas or misplaced modifiers, which could be improved. It might be helpful to have someone proofread the text to catch these issues. For instance, in some instances, it might be helpful to include a comma after introductory phrases for better readability (e.g., "In summary, ingesting NaHCO3 capsules...").
5. It would be beneficial to maintain consistency in verb tenses, especially when describing the study's methodology and findings. There are a few instances where the tense shifts unexpectedly, which could potentially disrupt the flow.

Response: Thank you for these grammatical suggestions. We have now proofread the manuscript and attempted to improve the readability throughout.

Reviewer 2 Report

Comments and Suggestions for Authors

Interesting, well conducted and written study.  I have a few comments for your consideration.

Line 19-20: Please explain why you gave placebo at 90 min and bicarb at 150 min?  Why not both at 150 min?

Line 96:  Please indicate why 5 men withdrew.  Was it related to the treatments?  Why not include the male who completed the study (you are eliminating almost 20% of your data) or at least discuss whether his responses differed from the women. 

Line 108: .... took place at the ....

Figures:  Please use open and closed symbols in addition to differing shapes to better discriminate treatments.  Your x axis [time (min)] needs to drawn to scale as non-scale presentations can mislead by altering the shape of the observed responses. 

Line 302. 72 should be 2.

Conclusion:  Your conclusion is a Summary and Conclusion not a conclusion. 

Author Response

Line 19-20: Please explain why you gave placebo at 90 min and bicarb at 150 min?  Why not both at 150 min?

Response: This was an attempt to blind the swimmers of their experimental condition. They were told they would be ingesting a supplement at three different timings (now made clear in lines 134 – 136), and were blind to both the individualised and standardised timings that would be used. We only gave pre-exercise ingestion timings to the swimmers in the week before each experimental trial. Standardised timings of 90 min and 150 min were selected as these strategies are currently recommended for athletes (Maughan et al., 2018), and did not clash with either of the swimmers’ individualised timings. We chose to use a placebo at 90 min pre-exercise since this timing may be too short to elicit peak blood bicarbonate values when ingesting sodium bicarbonate in capsules (Jones et al., 2016; Newbury et al., 2021). We have attempted to make this reasoning clearer in lines 152 – 154 and lines 157 – 159.

Line 96:  Please indicate why 5 men withdrew.  Was it related to the treatments?  Why not include the male who completed the study (you are eliminating almost 20% of your data) or at least discuss whether his responses differed from the women. 

Response: The reason for the withdrawals was coincidental and not related to the treatments or study. Instead, this study was conducted soon after the return to sport following COVID-19 and only involved one swimming club. As such, we speculate that low motivation linked these withdrawals. Hence, after the first swimmer left the sport, their teammates/friends became less motivated and followed. The one male who remained joined the club after COVID-19 and was highly motivated to compete at the Commonwealth Games, therefore his performance times were superior to the female participants. This information has been included to increase transparency in lines 89 – 99.

However, while the addition of the male skewed standard deviations within performance data, the statistical outputs both with (n=7) and without (n=6) his data were almost identical. Since his exclusion did not affect the outcomes of the study, we decided that focusing on a much-needed female cohort was more beneficial. However, if you feel that his addition would strengthen the study power, his data can be included.

Figures:  Please use open and closed symbols in addition to differing shapes to better discriminate treatments.  Your x axis [time (min)] needs to drawn to scale as non-scale presentations can mislead by altering the shape of the observed responses. 

Response: Thank you for this suggestion. We believe this amendment to include open and closed symbols have made figures 1-4 easier to interpret

With regards to the x axis, this is difficult to draw to scale as it was intended to show the variables at significant time points rather than changes over minutes. For example, baseline blood values were taken at different times for each participant due to the individual bicarbonate timings. Whereas the pre- and post-time trial values were taken just 2-3 min apart. This presentation is in agreement with similar bicarbonate research (Boegman et al., 2020; Gough et al., 2021; 2023).

Line 108: .... took place at the ....

Line 302. 72 should be 2.

Response: These errors have now been amended, along with other grammatical changes throughout the manuscript to improve the readability.

Conclusion:  Your conclusion is a Summary and Conclusion not a conclusion. 

Response: The conclusion has now been reworded (lines 423 – 438).